# IncVGGT: Incremental VGGT for Memory-Bounded Long-Range 3D Reconstruction

**Keyu Fang**
Duke University
keyu.fang@duke.edu

**Changchun Zhou**[*]
Duke University
changchun.zhou@duke.edu

**Yuzhe Fu**
Duke University
yuzhe.fu@duke.edu

**Hai "Helen" Li**
Duke University
hai.li@duke.edu

**Yiran Chen**
Duke University
yiran.chen@duke.edu

## ABSTRACT

We present IncVGGT, a training-free incremental variant of VGGT that makes transformer-based 3D reconstruction feasible for long sequences in real-world applications. VGGT relies on dense global attention, which causes memory to grow quadratically and requires excessive computation, making it impractical for long-sequence scenarios. Even evolved streaming variants, such as StreamVGGT, still suffer from rapidly growing cache and latency. IncVGGT addresses these challenges from two orthogonal directions: (1) register and fuse overlapping frames into composite views, reducing duplicate tokens, and (2) history-side pruning retains only the top-$k$ most relevant/highest-scoring slots together with the most recent one, bounding cache growth. This incremental and memory-efficient design minimizes computation and memory occupation across arbitrarily long sequences. Compared to StreamVGGT, IncVGGT sustains arbitrarily long sequences with large efficiency gains (e.g., on 500-frame sequences, 58.5$\times$ fewer operators, 9$\times$ lower memory, 25.7$\times$ less energy, and 4.9$\times$ faster inference) while maintaining comparable accuracy. In addition, unlike existing baselines that directly run out of memory beyond 300 (VGGT)–500 (StreamVGGT) frames, IncVGGT continues to operate smoothly even on 10k-frame inputs under an 80GB GPU, demonstrating that our design scales to ultra-long sequences without hitting memory limits. These results highlight IncVGGT's potential for deployment in resource-constrained edge devices for long-range 3D scenarios.

## 1 INTRODUCTION

3D reconstruction from videos for long-range 3D space is a long-standing challenge with broad impact across vision and robotics (Zhuo et al., 2025). Applications span immersive VR/AR/XR experiences (Hong et al., 2024; Zheng et al., 2024), real-time robotics and manipulation (Sünderhauf et al., 2018), autonomous navigation (Geiger et al., 2012). In many of these settings, models must run under strict memory and compute limits on edge or mobile hardware (Howard et al., 2017), where repeatedly processing overlapping content in long or redundant streams quickly becomes impractical.

Feed-forward transformer backbones have recently demonstrated strong 3D performance without per-scene optimization (Wang et al., 2024b; Leroy et al., 2024). VGGT (Wang et al., 2025) is a prominent example that predicts depth, pose, point maps, and tracks in a single pass. However, its global self-attention scales quadratically with the total token length (Vaswani et al., 2017), and naive streaming still accumulates an ever-growing cache. In practice, both behaviors become bottlenecks on long sequences, limiting real-time use in VR/AR devices, robots, and vehicles.

Our key observation is: video streams exhibit substantial redundancy along two axes (Yoon & Choi, 2023). **Input-side redundancy**: Adjacent frames repeatedly cover the same regions, so incurring

---

[*]Corresponding author.

the full token cost at every step leads to redundant computation(Habibian et al., 2021). **History-side redundancy**: Once relevance is known, many cached keys/values contribute little to the next step (Adnan et al., 2024). We instantiate these insights in IncVGGT, which retains the backbone's feed-forward nature while eliminating redundancy from two orthogonal directions. **Input side**: We register and compose short windows into compact composite views using a simple span gate: when viewpoints are coherent, frames are aligned and merged (Brown & Lowe, 2007). When they drift, the window is bisected with one shared overlap frame to preserve stitchability. This eliminates pixel-level overlap before tokenization, reducing the token load that enters attention. **History side**: We bound the key–value cache by relevance rather than sequence length: only the top-$k$ historically important slots together with the most recent one are retained for the next step. The result replaces dense "all tokens vs. all history" interactions with a sparse "few tokens vs. few slots" computation, delivering large savings in memory and operators while preserving geometric stability.

The design targets real deployments of 3D/4D perception: long recordings, stabilized capture, telepresence, and robotics all benefit from processing each region once and carrying forward only what remains useful. The registration-and-composition step includes lightweight handling of warping artifacts (voids and seams) so that tokenization remains stable (Li et al., 2015). The cache rule is a minimal change to the attention layer (Saxena et al., 2024). No iterative optimization or heavy post-processing is required, and both components run online. In this paper, we have the following two main contributions:

- **Input-side registration and composition.** We align and merge short windows into composite views, collapsing overlapping regions so that the transformer processes far fewer tokens (Sec. 3.1).

- **Global-local cache pruning.** We retain only a fixed number of globally high-score slots plus the most recent one, converting a growing cache into a constant-size set (Sec. 3.2).

## 2 BACKGROUND

**Structure-from-Motion (SfM) and Multi-view Stereo (MVS).** Classical 3D reconstruction approaches are rooted in geometry-based pipelines such as structure-from-motion (SfM) and multi-view stereo (MVS) (Ozyeşil et al., 2017; Wang et al., 2024a). These methods detect and match sparse image features, estimate camera poses through bundle adjustment, and recover dense geometry using photometric consistency. While highly accurate and theoretically well-grounded, they rely on iterative optimization and careful initialization, which makes them computationally demanding and often brittle in challenging scenarios (Zhang et al.). As a result, conventional SfM/MVS systems struggle to scale to long video sequences or real-time applications, often requiring hours of optimization for a single sequence. This motivates learning-based feed-forward approaches that can offer faster inference.

**Learning-based 3D Reconstruction.** Neural representations such as Neural Radiance Fields (NeRF) (Mildenhall et al., 2020) introduced a new paradigm for scene reconstruction, but typically require per-scene optimization and heavy computation, limiting their use in large-scale or long-sequence settings. To overcome these limitations, recent feed-forward methods directly predict geometry and pose from input images. For example, DUSt3R and MASt3R estimate point maps or depth maps in a single pass, while extensions like CUT3R (Wang et al.) attempt to incorporate memory mechanisms for longer sequences. These approaches improve efficiency and generalization, but still struggle with scalability, as frames are largely processed independently without exploiting temporal redundancy. This gap has motivated transformer-based designs that process all frames jointly.

**Transformer-based 3D Vision**. The success of transformers in 2D vision (Dosovitskiy et al., 2021) has inspired their application to multi-view 3D reconstruction. A representative example is VGGT, which employs a pure transformer backbone to jointly predict depth, camera pose, point maps, and feature tracks within a single feed-forward framework. By treating all input views as tokens in a global self-attention mechanism, VGGT achieves state-of-the-art accuracy across diverse 3D benchmarks and demonstrates the potential of transformers as a foundation model for geometry understanding with strong cross-dataset generalization.

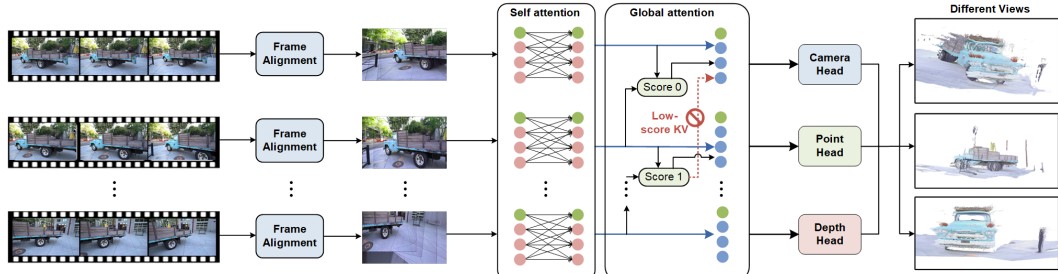

Figure 1: Overview of the IncVGGT pipeline. Input frames are aligned into a common reference before tokenization. Self- and global attention then operate with a bounded cache, where low-score keys/values are pruned. Outputs are predicted jointly by camera, point, and depth heads.

However, this design comes with significant scalability costs. The quadratic complexity of attention makes memory usage grow rapidly with the number of frames (Dao, 2023)—for instance, a 24GB GPU can only process a few dozen images before running out of memory. In addition, the model processes all frames jointly in one shot, which prevents incremental extension to longer sequences and causes resource usage to grow superlinearly as input length increases. As a result, VGGT achieves high accuracy on short sequences but cannot be directly applied to continuous or large-scale real-world videos. Addressing this limitation requires redundancy-aware strategies for both inputs and cached history.

**4D and Long-Sequence Reconstruction**. Extending transformer-based 3D models to long or continuous sequences has recently gained significant attention. VGGT-Long (Deng et al., 2025) tackles the scalability issue by dividing input videos into overlapping chunks and performing alignment across chunks, enabling large-scale reconstructions such as kilometer-long trajectories. However, this strategy is tailored for offline processing and requires additional post-hoc alignment. In contrast, StreamVGGT adapts VGGT to online video by introducing causal attention and key–value caching, allowing frames to be processed sequentially while reusing history rather than re-encoding all views jointly. StreamVGGT preserves the feed-forward nature of VGGT, supports real-time streaming inference, and currently represents the state of the art for long-sequence 4D reconstruction. Nevertheless, caching every key and value across time inevitably causes memory usage to grow with sequence length, so although StreamVGGT is effective in practice, its efficiency degrades over long videos, underscoring the need for redundancy-aware designs that scale gracefully with sequence length.

## 3 METHOD

We pursue scalability on high-overlap visual streams by removing redundancy at two complementary stages. First, an input-side registration and composition module collapses pixel-level overlap within a short window, yielding a compact composite views whose token count we denote by $\tilde{T}$ (Zitová & Flusser, 2003). Second, a global-local cache pruning module retains only $S$ historical slots (high-scoring plus recency) for attention, so both compute and memory remain controlled as sequences grow (Xiao et al., 2024; Kwon et al., 2023). Throughout we use $L$ for the total number of tokens in a dense global formulation; to avoid overloading notation, the span gate in Sec. 3.1 uses a separate threshold $\lambda$. Fig. 1 gives an overview of the pipeline.

### 3.1 REGISTRATION-BASED REDUNDANCY REDUCTION

Adjacent frames in long video streams often contain heavy pixel-level overlap, so processing each independently causes the transformer to tokenize the same regions multiple times. Our method mitigates input redundancy before attention by registering and composing a window of $K$ frames into a single composite view in a reference domain (Huang et al.). Registration-and-composition pipeline is illustrated in Fig. 2. We adopt a hierarchical composition strategy controlled by a span gate: first attempt to align and merge all $K$ frames; if the normalized coverage of the warped supports remains within a threshold $\lambda$, we generate one composite view; otherwise, the window is recursively bisected into two sub-windows until each satisfies the span criterion. To ensure frames remain

stitchable, adjacent sub-windows share one overlap frame. By collapsing overlapping pixels, the transformer is ultimately fed with a token count that scales with the composed support area rather than the raw number of frames. A statistical comparison of token growth under the baseline and our method is presented in Fig. 3.

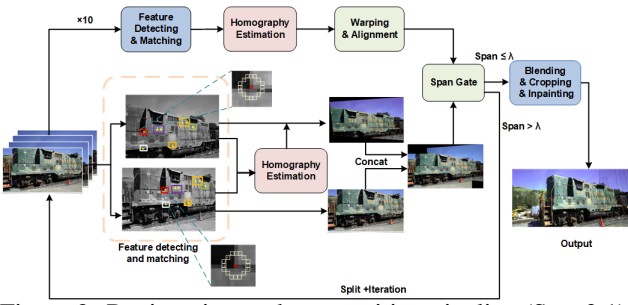

Figure 2: Registration-and-composition pipeline (Sec. 3.1). A short window of frames is aligned via feature matching and homography, warped into a reference domain, and composed through the span gate (split/merge) with lightweight blending/cropping/inpainting to produce a compact composite view.

Figure 3: Input token count vs. input size ($N$ denotes tokens of one standard view). Results are from `tandt_db truck` with our method.

**Feature detection and registration.** Given a short window of $K$ frames $\{I_i\}_{i=0}^{K-1}$, we first align all frames to a common reference before composition. Specifically, we select $I_0$ as the reference, extract local features (e.g., ORB (Rublee et al., 2011) or SIFT (Lowe, 2004)) on each frame, and establish candidate correspondences via k-nearest neighbors (kNN) matching. To reduce mismatches, we apply ratio tests and cross-checking, retaining only high-confidence correspondences, thus obtaining homographies $H_{i\to i-1}$ between adjacent frames (e.g. via DLT + RANSAC) (Agarwal et al., 2005). From these correspondences, we estimate pairwise homographies $H_{i\to i-1}$ that map frame $i$ to frame $i-1$. To propagate alignment to the reference, cumulative warps are obtained by composition:

$$H_{i\to 0} = H_{1\to 0}\, H_{2\to 1} \cdots H_{i\to i-1}, \qquad i = 1, \ldots, K-1, \tag{1}$$

with $H_{0\to 0} = I$. This procedure provides a lightweight yet effective alignment that tolerates outliers and mild parallax, while keeping all frames within a consistent reference domain (see the left block of Fig. 2). Although homographies cannot fully account for strong 3D parallax, this approximation proves sufficient for the short temporal windows considered. Notably, it allows efficient streaming implementation: once $H_{i\to i-1}$ is estimated, subsequent frames can be incrementally aligned to the reference without reprocessing earlier frames.

**Band-limited warping.** Direct projection of all warped frames onto a global canvas can cause the canvas size to grow unbounded, producing excessive extrapolation and large unsupported regions at the periphery. Such instability not only wastes tokens but also disrupts the subsequent span computation. To address this, we restrict composition to a narrow vertical band centered at the reference support. Concretely, a global translation is applied so that all warped coordinates remain nonnegative within this band-limited canvas. This design bounds the effective canvas size, suppresses unsupported margins, and ensures that span evaluation in the next stage remains well-conditioned.

**Normalized span and recursive bisection.** To decide whether $K$ frames can be composed in one shot, we define a normalized span that measures the horizontal coverage of the union of warped supports in units of a single-frame width. Let $\mathcal{S}_i = H_{i\to 0}(\Omega_i)$ be the warped support of frame $i$ (restricted to the band-limited canvas) and let $W_0$ be the width of $\Omega_0$. Using $B_x(\cdot)$ for the horizontal projection (the minimal interval covering a set along the $x$-axis),

$$\text{span}(\{H_{i\to 0}\}) = \frac{\left\lfloor B_x\left(\bigcup_{i=0}^{K-1}\mathcal{S}_i\right)\right\rfloor}{W_0}. \tag{2}$$

If span $\leq \lambda$, viewpoints are coherent and we proceed with single-shot composition. Otherwise we split into two sub-windows of size $\lfloor K/2 \rfloor$ and $\lceil K/2 \rceil$ that share the middle frame (e.g., $[0:\lfloor K/2 \rfloor]$

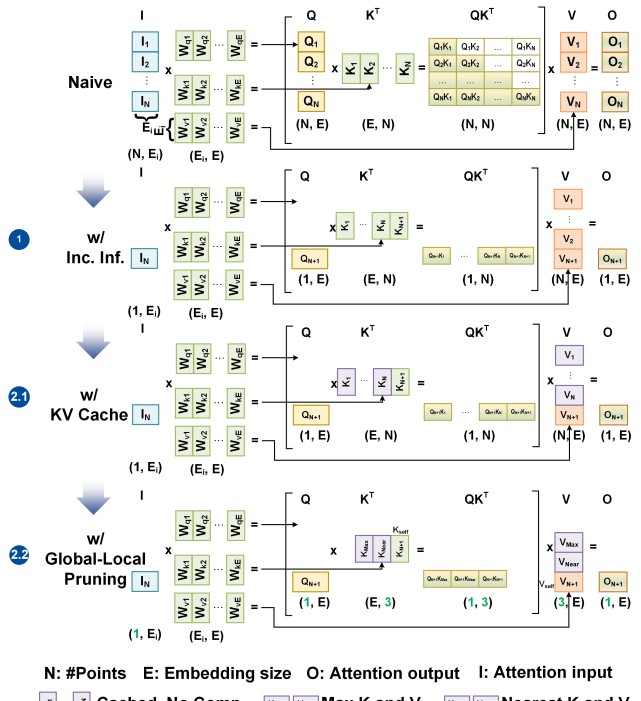

Figure 4: Top: **Naive** recomputation, where each new frame attends to all tokens ($L \times L$). 1: **Incremental inference**, which updates only the queries from the new frame. 2.1: **KV caching**, which reuses stored keys/values so new queries attend directly to the accumulated history. 2.2: **IncVGGT**, which selects only the top-$k$ maximum slots plus the most recent one, yielding a constant-size cache.

and $[\lfloor K/2 \rfloor{:}K]$), and recurse independently. The overlap frame serves as an alignment anchor that guarantees stitchability if higher-level fusion is needed; in practice, span decreases quickly with window size under typical motions.

**Blending and void mitigation.** Once a (sub-)window passes the span gate, its warped frames are fused by mask-aware distance-transform feathering (Allène et al., 2008). For each warp, we compute a binary validity mask and assign per-pixel weights that increase smoothly toward the interior, so that the normalized accumulation

$$\hat{I}(x) \;=\; \frac{\sum_i W_i(x)\,\tilde{I}_i(x)}{\sum_i W_i(x) + \delta}, \quad \delta > 0,$$

suppresses seams without introducing halo artifacts. To mimic streaming input, later frames overwrite earlier ones, ensuring that newly arrived content fills small residual gaps.

Planar warps may still produce unsupported voids, especially at disocclusions or near band boundaries. We apply a lightweight clean-up that operates only in these regions: persistent unsupported margins at the horizontal extremes are trimmed, and strict voids (all-zero pixels) are inpainted with a small-radius fill to remove speckles while preserving valid texture (Telea, 2004; Quan et al., 2024). Optionally cropping to the reference-frame height further regularizes the token grid and improves batching efficiency. The output examples in Fig. 2 illustrate typical seam and void artifacts together with their removal.

## 3.2 GLOBAL-LOCAL CACHE PRUNING

Simply storing all past keys and values, as done in vanilla streaming transformers (Dai et al., 2019), leads to ever-growing cache size and rising cost, making long sequences infeasible. In VGGT, each incoming frame triggers a full recomputation of global attention across all tokens, producing a dense $L \times L$ matrix where $L = F \cdot p$ is the total token length for $F$ frames with $p$ tokens per frame (Fig. 4, top). This quadratic cost requires both heavy computation and memory to maintain

the key–value states. Incremental inference mitigates this by processing only the queries from the new frame rather than recomputing all past ones (Fig. 4, 1). KV caching further improves efficiency by storing historical keys/values so that new queries can directly attend to them (Fig. 4, 2.1) (Zhao et al., 2024). However, the per-step cost $\mathcal{O}(B \cdot H \cdot p \cdot L_{\text{hist}} \cdot d_h)$ still grows linearly with cache length. Once the cache accumulates 7–800 frames, memory already exceeds 80 GB on an A100 and per-frame inference time keeps rising with sequence length. This motivates a more selective strategy: instead of retaining all slots, we keep only a compact set of globally high-score slots plus the most recent one, while low-score slots—often arising from overlapping or repetitive views—are pruned as redundant (Fig. 4, 2.2).

**Global-local pruning rule.** Real-world video streams for 3D reconstruction exhibit strong frame-to-frame continuity and overlap. Slots with the highest scores at step $t$ are likely to be reused at step $t+1$. We exploit this property by propagating historical relevance and preselecting only the top-$k$ slots, together with the most recent one to capture sudden scene changes (Kim & Jung, 2025; Fu et al., 2024). Formally, let

$$A_t = \text{softmax}\left(\frac{Q_t K_{1:L_{\text{hist}}}^\top}{\sqrt{d_h}}\right) \tag{3}$$

be the attention over cached keys at time $t$. Reducing $A_t$ across queries/heads yields a per-key relevance vector $s^{(t)} \in \mathbb{R}^{L_{\text{hist}}}$. Before processing frame $t+1$, we preselect

$$\mathcal{S}_{t+1} = \text{TopK}\left(s^{(t)}, k\right) \cup \{\text{most recent}\},$$

and restrict attention to this $(k+1)$-sized set:

$$O_{t+1} = \text{softmax}\left(\frac{Q_{t+1} K_{\mathcal{S}_{t+1}}^\top}{\sqrt{d_h}}\right) V_{\mathcal{S}_{t+1}}. \tag{4}$$

This avoids scoring against the entire cache and yields constant-size history per step.

**Complexity and memory.** Restricting attention to $\mathcal{S}_{t+1}$ bounds the per-step score computation to $\mathcal{O}(B \cdot H \cdot (k+1) \cdot d_h)$ and the KV footprint to $\mathcal{O}((k+1) \cdot d_h)$ per head, both independent of sequence length. Over $T$ steps, the cost is linear in $T$ for fixed $k$, while retaining enough temporal context due to inter-frame redundancy.

**Combined effect.** Let $S$ denote the number of retained historical tokens ($S = k+1$) and $\tilde{T}$ the composed token count from Sec. 3.1. The dense baseline over $L$ tokens costs $\mathcal{O}(B \cdot H \cdot L^2 \cdot d_h)$ per layer. With registration reducing inputs to $\tilde{T}$ and the cache limited to $S$, the per-step cost becomes $\mathcal{O}(B \cdot H \cdot \tilde{T} \cdot S \cdot d_h)$ and stored key–value memory is bounded by $\mathcal{O}(S \cdot d_h)$ per head. Together, input-side registration and history-side pruning convert quadratic growth into much smaller $\tilde{T} \cdot S \ll L^2$, significantly reducing both operator count and memory footprint at the method level.

# 4 EXPERIMENTS

## 4.1 EXPERIMENTAL SETUP

All experiments are conducted on a single NVIDIA A100 GPU (80GB). The input image height is fixed at 518 pixels while preserving the original aspect ratio, ensuring consistent resolution across datasets. During inference, inputs are constructed as sliding windows of 10 consecutive frames, with each prediction aligned to the first frame of the window. For history compression, we retain the top-$k = 5$ cached frames, and for multi-view selection, the top-20 camera tokens, corresponding to roughly five neighboring views. Unless otherwise specified, models are evaluated using official StreamVGGT weights with our inference modifications. Efficiency is measured in terms of inference latency, GPU memory (allocated and reserved), GEMM GFLOPs, and hardware-level metrics including average power and energy per output. Accuracy is assessed on multi-view reconstruction and long-sequence depth estimation tasks.

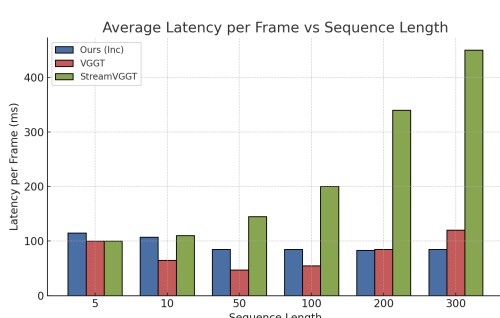

Figure 5: Per-frame latency vs. sequence length on tandt$_d$ band$KITTI$ datasets.

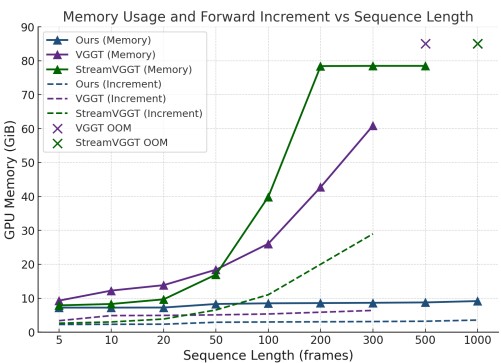

Figure 6: Reserved and incremental memory vs. sequence length. OOM indicates out-of-memory.

Table 1: Inference time (s) across different sequence lengths (OOM indicates out-of-memory).

| Method | 5 | 10 | 20 | 50 | 100 | 200 | 300 | 500 | 1k | 10k |
|---|---|---|---|---|---|---|---|---|---|---|
| Ours (Inc) | 0.57 | 1.04 | 2.09 | 4.17 | 8.20 | 16.28 | 25.50 | 38.12 | 82.00 | 797.31 |
| StreamVGGT | 0.50 | 1.06 | 2.15 | 7.23 | 20.09 | 68.22 | 136.72 | 185.11 | OOM | OOM |
| VGGT | 0.51 | 0.63 | 1.09 | 2.37 | 5.78 | 17.34 | 36.32 | OOM | OOM | OOM |

## 4.2 EFFICIENCY EVALUATION

**Latency.** Latency is evaluated on Tanks-and-Temples (train and truck) for sequences up to 300 frames, and on KITTI for longer sequences (Knapitsch et al., 2017). Per-frame latency is computed as total inference time divided by sequence length; for sequences longer than 20 frames, both the merging step and the forward pass are included. Figure 5 shows per-frame latency (ms/frame), while Table 1 indicates total sequence time (s). Our method remains nearly constant once the sequence exceeds 100 frames, whereas StreamVGGT rises sharply and VGGT also grows with length. For example, from 10 to 300 frames, our method decreases slightly from about 104 ms to 85 ms per frame, while StreamVGGT increases from 106 ms to 455 ms and VGGT from 63 ms to 117 ms. This demonstrates that our incremental design successfully prevents latency per frame from scaling with sequence length.

At the sequence level, Table 1 shows the same trend. At 100 frames, our method completes in 8.20 s, compared to 20.09 s for StreamVGGT (2.4× faster) and 5.78 s for VGGT. At 300 frames, the difference is substantial: our method finishes in 25.54 s, whereas StreamVGGT requires 136.72 s (5.4×), and VGGT 35.18 s. These results highlight that our incremental design delivers substantially faster inference than StreamVGGT, while maintaining comparable accuracy to VGGT and even surpassing it in some cases.

**Memory Usage.** Since our focus is on memory bottlenecks, we present reserved memory throughout this section, as it directly reflects the peak footprint that determines CUDA out-of-memory (OOM) failures. Here, OOM denotes the runtime termination when the GPU cannot allocate further memory. We further define incremental memory as the additional reserved usage incurred per forward step. As shown in Figure 6, both VGGT and StreamVGGT exhibit steep growth in reserved memory with longer sequences. VGGT already exceeds 60 GB at 300 frames and fails thereafter, even without bundle adjustment. StreamVGGT behaves similarly, reaching more than 78 GB and crashing at 1k frames. In contrast, our method maintains a nearly flat reserved memory profile across all tested lengths, staying below 9 GB even at 1k frames. This stability indicates that our redundancy-aware state compression effectively breaks the quadratic growth trend that makes existing approaches impractical for long sequences.

We further examine incremental memory, defined as the additional reserved usage incurred per forward step. Again in Figure 6, both baselines show rapidly increasing increments, with VGGT and StreamVGGT surpassing 5–20 GB once the input length exceeds 100 frames. Our method, however, keeps the incremental cost tightly bounded around 2–3 GB regardless of sequence length. This

Table 2: Theoretical attention operator counts (in k) computed on the KITTI dataset.

| #Frames | VGGT | StreamVGGT | Ours (Inc) |
|---|---|---|---|
| 5 | 7.19k | 6.38k | 6.38k |
| 50 | 286.52k | 163.42k | 11.05k |
| 100 | 951.58k | 548.13k | 27.39k |
| 200 | 3603.63k | 1981.39k | 70.26k |
| 300 | 7956.14k | 4299.78k | 126.33k |
| 500 | OOM | 11591.97k | 198.88k |

Table 3: Energy consumption (J) across different sequence lengths. The last column presents the average power consumption (W).

| Method | 5 | 20 | 50 | 100 | 200 | 300 | 500 | 1k | Avg Power (W) |
|---|---|---|---|---|---|---|---|---|---|
| Ours (Inc) | 64.7 | 238.0 | 113.7 | 202.8 | 455.8 | 811.0 | 1183.2 | 2885.3 | 135.7 |
| VGGT | 62.2 | 105.8 | 513.4 | 1269.3 | 3926.9 | 8078.7 | OOM | OOM | 178.3 |
| StreamVGGT | 125.6 | 309.2 | 1119.2 | 3360.9 | 11228.2 | 23516.1 | 30503.5 | OOM | 153.8 |

bounded slope is critical: it ensures that even under long streaming inputs, memory growth remains predictable and sustainable. Together, these findings demonstrate that our approach is not only efficient in terms of total footprint but also robust in scaling behavior, enabling reliable operation on GPUs where existing models would immediately run out of memory. More broadly, the combination of low reserved footprint and bounded incremental cost highlights that our method is well-suited for edge cases and commodity GPUs, where both absolute capacity and memory headroom are highly constrained.

**Computation efficiency.** To further analyze computational efficiency, we estimate the number of attention operators, which contribute over 95% of the total FLOPs (Sreedhar et al., 2022). We present theoretical values derived from matrix dimensions, focusing only on the attention component. As shown in Table 2, our method reduces operator counts dramatically compared to VGGT and StreamVGGT. For example, at 300 frames, VGGT requires about 7,956k attention operators and StreamVGGT about 4,300k, whereas our method uses only 126k—over 60× fewer than VGGT and 30× fewer than StreamVGGT. This significant reduction directly explains the improvements in latency and energy, since far fewer redundant attention computations are executed while reconstruction quality is preserved.

**Energy Efficiency.** Beyond memory and latency, we further examine hardware-level efficiency in terms of energy consumption (Cao et al., 2021). As summarized in Table 3, our method consistently requires substantially less energy than both VGGT and StreamVGGT across all sequence lengths. While the average power draw is slightly lower than competing baselines (135.7 W vs. 153.8 W and 178.3 W), the key advantage comes from significantly shorter runtimes, which translate into much lower energy-per-sequence. For instance, at 300 frames our method consumes only 811 J, compared to 8079 J for VGGT and over 23,000 J for StreamVGGT. This one-order-of-magnitude gap widens as the sequence length increases. Since all methods are evaluated on the same sequence lengths, total energy is directly comparable; equivalently, dividing by frame count yields consistent J/frame trends. These results indicate that our redundancy-aware design not only conserves memory but also reduces wasted GPU cycles. As a result, this energy efficiency makes the approach especially appealing for resource-constrained settings, such as commodity GPUs or edge devices, where both memory and power budgets are limited. In practice, this means our model can sustain long-sequence reconstruction tasks on hardware platforms where other methods would either exhaust memory or exceed acceptable power envelopes, broadening the applicability of foundation models in 3D vision.

## 4.3 ACCURACY EVALUATION

We evaluate accuracy on video depth and multi-view reconstruction, as single-frame and camera pose estimation are nearly identical to StreamVGGT due to shared pretrained weights. Our goal is therefore to confirm that the efficiency gains described in Section 4.2 are achieved while maintaining accuracy at a comparable level, without incurring significant degradation.

Table 4: Video depth evaluation on Sintel, Bonn, and KITTI datasets. Results for StreamVGGT and ours are obtained from our own evaluation, while others are derived from StreamVGGT. $\delta<1.25$ is reported as a percentage.

| Method | Type | Sintel | | Bonn | | KITTI | |
|---|---|---|---|---|---|---|---|
| | | Abs Rel $\downarrow$ | $\delta<1.25\uparrow$ | Abs Rel $\downarrow$ | $\delta<1.25\uparrow$ | Abs Rel $\downarrow$ | $\delta<1.25\uparrow$ |
| VGGT | Dense-view | 0.298 | 68.1 | 0.057 | 96.8 | 0.061 | 97.0 |
| Spann3R | Streaming | 0.622 | 42.6 | 0.144 | 81.3 | 0.198 | 73.7 |
| CUT3R | Streaming | 0.421 | 47.9 | 0.078 | 93.7 | 0.118 | 88.1 |
| Point3R | Streaming | 0.452 | 48.9 | 0.060 | 96.0 | 0.136 | 84.2 |
| StreamVGGT | Streaming | 0.328 | 64.9 | 0.059 | 97.2 | 0.173 | 72.2 |
| Ours (Inc) | Streaming | 0.341 | 63.1 | 0.064 | 97.0 | 0.176 | 71.9 |

Table 5: Multi-view reconstruction on 7Scenes and NRGBD. Include median accuracy (Acc), completeness (Comp), and normal consistency (NC1).

| Method | 7Scenes | | | NRGBD | | |
|---|---|---|---|---|---|---|
| | $Acc_{med}\downarrow$ | $Comp_{med}\downarrow$ | $NC1_{med}\uparrow$ | $Acc_{med}\downarrow$ | $Comp_{med}\downarrow$ | $NC1_{med}\uparrow$ |
| VGGT | 0.0055 | 0.0067 | 0.948 | 0.0139 | 0.0147 | 0.993 |
| StreamVGGT | 0.0241 | 0.0203 | 0.915 | 0.0520 | 0.0340 | 0.988 |
| Ours (Inc) | 0.0266 | 0.0203 | 0.901 | 0.0516 | 0.0345 | 0.987 |

**Video Depth.** Table 4 shows video depth results on Sintel (Butler et al., 2012), Bonn (Palazzolo et al., 2019), and KITTI. Across datasets, our accuracy remains very close to StreamVGGT, with relative differences typically within 1–3%. For example, on Sintel the Abs Rel differs by only about 4%, and on KITTI the gap is below 2%. On Bonn, both methods achieve nearly identical $\delta<1.25$ scores around 97%. These small gaps indicate that the efficiency gains of our method are realized with only minimal loss in accuracy, while still outperforming other streaming baselines such as Spann3R, CUT3R, and Point3R by a clear margin.

**Multi-view Reconstruction.** Table 5 shows results on 7Scenes (Shotton et al., 2013) and NRGBD (Azinović et al., 2022). VGGT achieves the lowest errors overall, but it is a non-streaming baseline and cannot scale beyond short sequences. Among streaming models, the accuracy (Acc) metric shows complementary strengths: on 7Scenes, StreamVGGT attains slightly lower Acc, while on NRGBD our method achieves marginally better Acc. Other metrics remain largely comparable, and in all cases the differences are within 0.3%, indicating that our approach matches the reconstruction quality of the best streaming baseline, while uniquely offering the efficiency and scalability demonstrated in Section 4.2.

## 5 DISCUSSION

We focus on a training-free, redundancy-aware inference framework that remains efficient in extremely long sequences. This design offers strong scalability and memory savings but also introduces a natural trade-off. When the overlap ratio becomes very small or decreases rapidly, such as in scenes with large viewpoint changes or limited texture, the reduced geometric redundancy can influence the stability of the registration module. We also discuss two possible modes for long-sequence reconstruction: training-free and finetuning. We choose the training-free mode because IncVGGT already preserves a comparably high level of accuracy, leaving limited room for finetuning to produce substantial gains, and because the training-free setting keeps the method architecture-agnostic and directly applicable to arbitrarily long inputs. Finetuning remains a compatible option through existing VGGT and StreamVGGT pipelines, but it is orthogonal to the core focus of this work. These choices reflect practical compromises that allow the system to stay lightweight and highly memory efficient.

## 6 CONCLUSION

We presented IncVGGT, an incremental and memory-efficient visual geometry transformer for long-range 3D reconstruction. Our design collapses redundant input views into compact composites

and prunes the history cache with a top-$k$-plus-recency rule. This ensures both operator count and memory footprint remain bounded while preserving awareness of global and local spatial context. Experimental results show that IncVGGT outperforms state-of-the-art methods by nearly an order of magnitude. IncVGGT scales effectively to long sequences while preserving accuracy, avoiding the memory and latency limitations of existing baselines. More broadly, IncVGGT demonstrates that redundancy-aware design can make transformer-based reconstruction practical for VR/AR, robotics, and edge devices, enabling reliable deployment in real-world long-sequence scenarios.

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

## A ADDITIONAL QUALITATIVE VISUALIZATIONS

Figures 7, 8, and 9 provide additional qualitative comparisons between the proposed IncVGGT, the prior StreamVGGT, and VGGT-Long on long-sequence reconstruction (On KITTI dataset). As shown, IncVGGT produces more complete scene geometry with clearer high-frequency structures such as trees, lamp posts, and roadside details. This improvement mainly comes from reducing the large amount of near-duplicate frames present in real long streams: our registration module merges these highly redundant views, which reduces conflicting signals, prevents the backbone from generating scattered outliers, and avoids the loss of geometry that can occur when the KV-cache becomes overloaded.

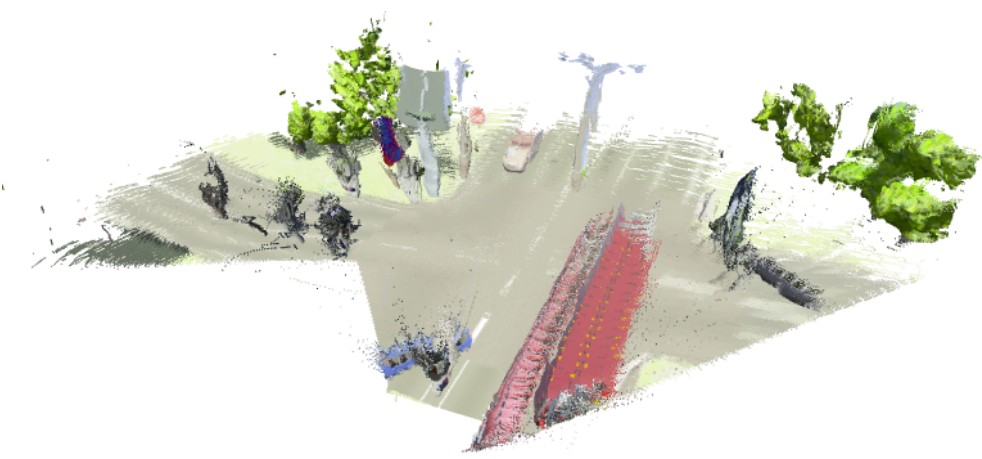

Figure 7: Qualitative reconstruction produced by **IncVGGT**.

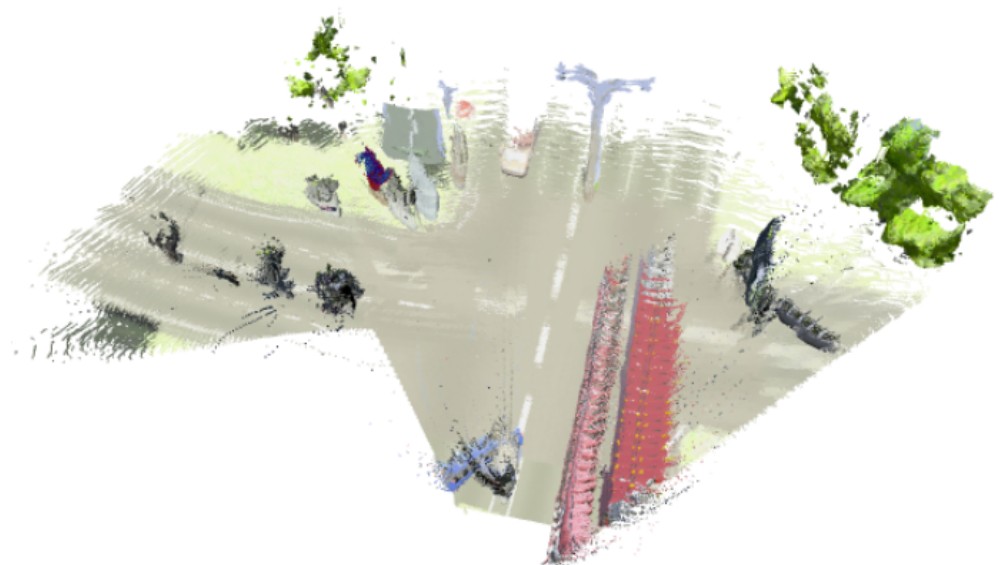

Figure 8: Reconstruction produced by **StreamVGGT**.

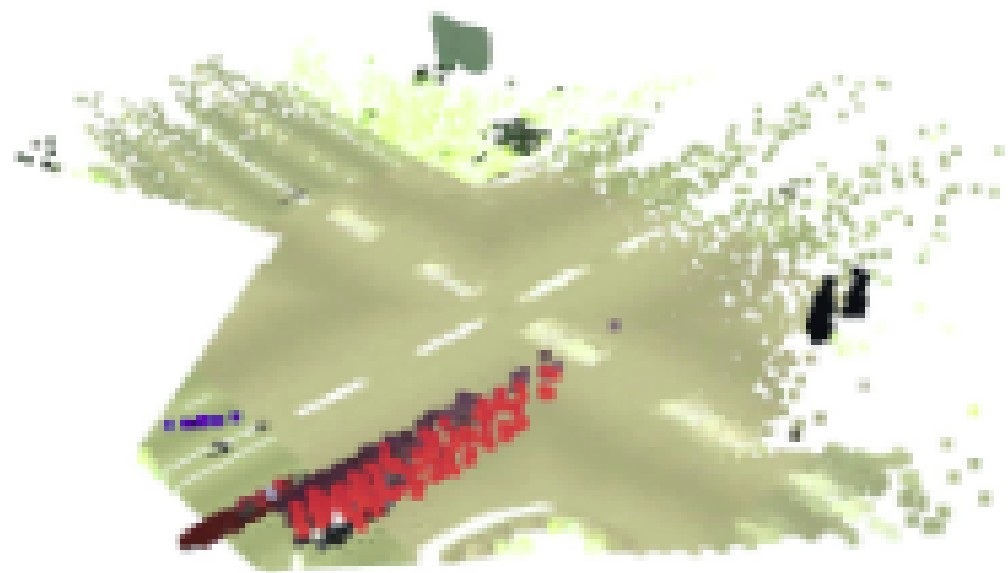

Figure 9: Reconstruction produced by **VGGT-Long** (The visualization is taken from the original VGGT-Long paper).

