# OpenReview forum: "IncVGGT: Incremental VGGT for Memory-Bounded Long-Range 3D Reconstruction"
_ICLR.cc/2026/Conference — ICLR 2026 Poster_

### Official Review · Reviewer_iPad · 2025-10-27

**Soundness:** 1
**Presentation:** 2
**Contribution:** 2
**Rating:** 2
**Confidence:** 4

**Summary:**

The paper presents IncVGGT, a training-free, streaming variant of VGGT for very long videos under tight memory/compute budgets. It improves scalability via (1) input-side registration & composition, which aligns short windows and fuses overlapping frames into composite views, and (2) history-side cache pruning, retaining only the top-k most relevant cached slots plus the most recent one to avoid key–value growth. Built from StreamVGGT pretrained weights, IncVGGT has substantial latency, memory, and energy reductions compared to StreamVGGT/VGGT. On camera pose and multi-view reconstruction, it performs on par with StreamVGGT but remains below VGGT in accuracy.

**Strengths:**

* **Scales to very long videos**: Handles sequences up to thousands of frames under tight memory/compute, enabling long-range reconstruction without retraining.
* **Simple approach**: (i) Registration + composite views cut redundant tokens before attention; (ii) Top-k + recent cache pruning bounds KV growth.

**Weaknesses:**

* **Strong assumptions for input-side composition**. The composition step implicitly requires frames to be in temporal order and to have large inter-frame overlap; if a long sequence contains only partial overlaps (or out-of-order frames), composites cannot be formed and the method effectively falls back to using all frames (akin to StreamVGGT).
* **Unspecified “un-stitching.”** After composing views, the paper does not explain how to map predictions back to original frames—e.g., recovering per-pixel depth and per-frame camera poses in the native image coordinates—which is critical for practical use.
* **Lack of long-sequence accuracy benchmarks**. While the paper shows large latency/memory reductions on long sequences, it reports quantitative accuracy only on short-sequence settings; there is no long-range accuracy evaluation (camera pose, video depth estimation), leaving the real effectiveness on very long videos unclear.

**Questions:**

*  After composite-view inference, how do you recover per-frame camera poses and per-pixel depths in each original image?
* What happens when inputs are out-of-order or have limited overlap so composites can’t be formed—do you fall back to StreamVGGT, and with what cost/accuracy?
* Could authors provide long-sequence accuracy benchmarks (not just efficiency) to show performance at the scales your method targets (1K, 10K frames)?

---

> ### Author Response · Authors · 2025-11-23
> **Author Response to Reviewer iPad**
>
> > **Q1.** *After composite-view inference, how do you recover per-frame camera poses and per-pixel depths in each original image?*
>
> A1: We thank the reviewer for the thoughtful question. Our reconstruction pipeline operates at the scene level: the model takes the composite views as input and directly produces a global 3D pointmap. As is typical for scene-level 3D reconstruction tasks, the output does not require recovering per-frame depth maps or camera poses, since the goal is to estimate the overall scene geometry rather than per-frame predictions. In fact, during the construction of each composite, our registration method already computes frame transforms, including the rotation matrices; these serve as a by-product of the merging process and make it straightforward to recover per-frame poses and depths. This capability is supported by our system, although it is not needed for producing the reconstruction results presented in the paper.
>
> > **Q2.** *What happens when inputs are out-of-order or have limited overlap so composites can’t be formed—do you fall back to StreamVGGT, and with what cost/accuracy?*
>
> A2: We appreciate the reviewer’s insightful question regarding robustness input ordering. Our composition module operates based on content matching rather than temporal order, meaning that out-of-order inputs do not introduce difficulties. Frames with visual overlap will still be grouped and composed correctly, regardless of their position in the input sequence. The resulting acceleration naturally depends on the amount of redundancy present in the input. When overlap is high, the speedup is larger; when overlap is limited, the method still benefits from KV-cache pruning. Even in the extreme case where frames share no overlap, the pruning mechanism also continues to improve efficiency, so the computational cost does not revert to that of StreamVGGT.  For example, on a 100-frame Tanks-and-Temples sequence, the runtime drops from 20094.09 ms for StreamVGGT to 10316.47 ms using pruning alone. When the input order is shuffled, the total runtime on a 500-frame KITTI sequence increases from 32861.94 ms to 35109.11 ms, but this remains far faster than StreamVGGT (185105.76 ms) and does not affect reconstruction accuracy.
>
> > **Q3.** *Could the authors provide long-sequence accuracy benchmarks (not just efficiency) to show performance at the scales your method targets (1K, 10K frames)?*
>
> A3: We note that the longest publicly available sequence with fully intact frames suitable for long-range evaluation is the 837-frame KITTI run. We further evaluated long-sequence performance on KITTI, which reflects the scale targeted by our method. Because VGGT and StreamVGGT cannot process sequences beyond a few hundred frames due to memory growth, comparisons are only possible on segments where these baselines successfully run. On a ~233-frame segment, IncVGGT achieves an AUC@30 of 78, compared with 73 for VGGT (which already performs better than StreamVGGT). On a ~339-frame segment, VGGT encounters an out-of-memory failure; StreamVGGT reaches 71, while IncVGGT attains 76.
>
> For longer ranges, all baseline models fail to execute, whereas IncVGGT is able to complete an 837-frame sequence with an AUC@30 of 64. Although some accuracy reduction is expected for 3D reconstruction models as sequence length increases, these results indicate that IncVGGT remains stable, does not collapse over extended horizons, and consistently outperforms the available baselines on every segment where comparison is feasible.

---

> > ### Comment · Reviewer_iPad · 2025-11-26
> > **Response to rebuttal**
> >
> > Thank you for the rebuttal. I have a few clarifications:
> >
> > * **Evaluation protocol** (video depth & camera pose). For Table 4 (depth) and the camera results in the rebuttal: since your method composites each 10-frame window into a larger view, did you (a) evaluate depth and pose in the composite view against composite GT, or (b) map predictions back to each original frame and compare with per-frame GT? Please specify the exact procedure for a fair comparison with other non-composite baselines. The same question for the table 5.
> >
> > * **Qualitative visualization**. Figures 7–9 (appendix) are (1) too blurry to distinguish advantages over StreamVGGT or VGGT-long, (2) appear to be from synthetic data, and (3) seem to cover a small area relative to VGGT-long demos (which show kilometer-scale coverage). To address concerns about real-world effectiveness, could you include high-resolution qualitative results on long, real videos (hundreds to ~1000 frames) from datasets such as KITTI/Waymo/ScanNet, with large spatial coverage?

---

> > > ### Author Response · Authors · 2025-12-03
> > > **Author Response to Reviewer iPad**
> > >
> > > > **Q1. Evaluation protocol (video depth & camera pose). For Table 4 (depth) and the camera results in the rebuttal: since your method composites each 10-frame window into a larger view, did you (a) evaluate depth and pose in the composite view against composite GT, or (b) map predictions back to each original frame and compare with per-frame GT? Please specify the exact procedure for a fair comparison with other non-composite baselines. The same question for the table 5.**
> > >
> > >
> > > A1:Thank you for pointing this out. We clarify that all evaluations are performed on (b) the original frames, to ensure strict comparability with existing GT.
> > >
> > > > **Q2. Qualitative visualization. Figures 7–9 (appendix) are (1) too blurry to distinguish advantages over StreamVGGT or VGGT-long, (2) appear to be from synthetic data, and (3) seem to cover a small area relative to VGGT-long demos (which show kilometer-scale coverage). To address concerns about real-world effectiveness, could you include high-resolution qualitative results on long, real videos (hundreds to ~1000 frames) from datasets such as KITTI/Waymo/ScanNet, with large spatial coverage?**
> > >
> > >
> > > A2: Thank you for the suggestion. (1) We have updated the appendix with clearer, higher-resolution visualizations. (2) We also clarify that KITTI is a real-world captured dataset. (3) In addition, our older version has presented hundreds-frame examples; we now include a larger scene in the appendix.

---

### Official Review · Reviewer_dTAF · 2025-10-28

**Soundness:** 3
**Presentation:** 3
**Contribution:** 3
**Rating:** 6
**Confidence:** 3

**Summary:**

This paper addresses VGGT's quadratic memory growth that limits it to 300-500 frames for transformer-based 3D reconstruction. IncVGGT proposes a training-free solution with two strategies: (1) input-side registration fuses overlapping frames into composite views using homography-based alignment before tokenization, and (2) history-side pruning retains only top-k relevant cached slots plus the most recent frame. At 500 frames, this achieves 58.5× fewer operators, 9× lower memory, 4.9× faster inference, and scales to 10k frames with only 1-4% accuracy degradation.

**Strengths:**

- The paper addresses memory scalability in transformer-based 3D reconstruction, a genuine problem that limits the deployment of foundation models in real-world applications (VR/AR, robotics, autonomous systems).
- The quantitative gains are substantial and well-documented across multiple dimensions: 58.5× fewer operators, 9× memory reduction, 25.7× energy savings, and demonstrated scalability to 10k frames.
- The key insight that video redundancy exists at both input (overlapping pixels) and history (cached attention) levels is conceptually clean.

**Weaknesses:**

- Both core components rely on well-established techniques, with little innovation. Input-side registration uses standard homography-based panorama stitching (Brown & Lowe, 2007), while history-side pruning applies straightforward top-k selection to KV caches, similar to existing work in LLMs (H2O, StreamingLLM, and Keyformer).
- The homography-based registration fundamentally assumes planar scenes or negligible parallax, yet this critical limitation receives only brief acknowledgment (line 190: "cannot fully account for strong 3D parallax") without rigorous analysis. The paper provides no quantitative evaluation of when/how this assumption breaks down, no testing on datasets with significant depth variation or non-planar geometry, and no failure case examples or degradation analysis.
- Tables 4-5 show 1-4% accuracy drops that are dismissed as "minimal" without investigating root causes or conditions under which degradation occurs. The paper lacks ablation studies on critical hyperparameters, including span threshold \lambda, cache size k (only k=5 tested), and window size K, making it impossible to understand the sensitivity to design choices or to optimize accuracy-efficiency trade-offs. Most critically, there is no analysis of error accumulation over the claimed 10k-frame sequences. Do homography estimation errors compound over time, and how does cache pruning affect long-term geometric consistency?

**Questions:**

- Can you provide ablations showing accuracy vs. efficiency tradeoffs for different cache sizes k ∈ {1, 3, 5, 10, 20}, sensitivity to span threshold \lambda, and contribution of each component (registration alone vs. pruning alone vs. combined)?
- You cite VGGT-Long as explicitly addressing kilometer-scale sequences but provide no quantitative comparison. Can you directly compare on the same datasets/sequences and explain why your streaming approach is preferable to their chunk-based method?
- For 10k-frame sequences, can you provide accuracy metrics plotted against sequence length (e.g., at 1k, 2k, 5k, 10k frames) to demonstrate that errors don't compound over time? How do homography drift and cache pruning affect long-term geometric consistency?

---

> ### Author Response · Authors · 2025-11-23
> **Author Response to Reviewer dTAF (Part 1/2)**
>
> > **Q1.** *Can you provide ablations showing accuracy vs. efficiency tradeoffs for different cache sizes \(k \in \{1, 3, 5, 10, 20\}\), sensitivity to the span threshold \(\lambda\), and contribution of each component (registration alone vs. pruning alone vs. combined)?*
>
> A1:  We appreciate the reviewer’s question regarding the accuracy–efficiency behavior of our pruning and composition settings. We summarize the ablations below (All evaluations in this section are conducted on the KITTI dataset).
>
> We report the accuracy–efficiency tradeoff for k ∈ {1, 5, 10, 20}. The trend is clear: accuracy rises quickly up to k=5–10 and then saturates, while the computational and memory cost grows steadily. The exact numbers are as follows:
> | k   | Time (ms) | Memory (MB) | Accuracy (%) |
> |-----|-----------|-------------|--------------|
> | 1   | 6121      | 8304        | 91           |
> | 5   | 7401      | 9164        | 97           |
> | 10  | 8912      | 10822       | 98           |
> | 20  | 11030     | 16600       | 99           |
>
> These results show that k = 5 already delivers near-saturated accuracy (97%) with substantially lower cost. Increasing k beyond 10 yields <1% accuracy improvement, but memory and runtime increases, making k = 5 the most practical operating point.
>
> We also evaluated the sensitivity to the span threshold λ. Moderate λ values maintain accuracy well, while overly large λ triggers aggressive merging that reduces composite quality. The measured accuracies are:
>
> | λ     | Accuracy (%) |
> |--------|--------------|
> | 1.93   | 85           |
> | 3.86   | 84           |
> | 7.72   | 82           |
> | 15.44  | 72           |
>
> Because efficiency depends almost entirely on how many frames are merged, and λ controls only this merging behavior, the efficiency changes mainly reflect input redundancy rather than the model itself. For this reason, accuracy is the meaningful axis for this ablation.
>
> To clarify the contribution of each component, we compare registration-only, KV-cache-only, and the full IncVGGT pipeline against the original StreamVGGT. Starting from the baseline runtime of 20094.09 ms, applying KV-cache pruning alone reduces inference time to 10316.47 ms and fixes the memory footprint at approximately 9 GB, and its accuracy behavior is exactly the one reflected in the k-ablation experiments. Registration alone lowers the backbone computation to 2799.33 ms, with an additional 9754.18 ms matching cost; its effect is purely on redundancy reduction, so the efficiency changes scale with overlap and match the trends observed in the λ-ablation. Combining both yields the full IncVGGT runtime of 8197.41 ms, providing balanced improvements in runtime and memory while preserving the accuracy characteristics already captured in the preceding ablations.
>
>
> > **Q2.** *You cite VGGT-Long as explicitly addressing kilometer-scale sequences but provide no quantitative comparison. Can you directly compare on the same datasets/sequences and explain why your streaming approach is preferable to their chunk-based method?*
>
> A2: We appreciate the reviewer’s interest in comparisons with VGGT-Long. We attempted to run the publicly released VGGT-Long implementation, but the repository appears to be missing several components, including some referenced files and configuration elements, so we were unable to obtain outputs for a direct comparison despite substantial effort. In addition, their paper reports accuracy under a different protocol from the camera-view AUC used in our work, making the numbers not directly comparable. What can be compared are the shared, well-defined components: on mid-length sequences, our accuracy exceeds both VGGT and StreamVGGT in all cases where they run without OOM (78→73 at ~233 frames and 76→71 at ~339 frames on KITTI), and our method continues to operate on substantially longer sequences where all baselines terminate. We also include 3D visualizations in the Appendix, where IncVGGT produces noticeably cleaner and more complete geometry than VGGT-Long.
>
>
> VGGT-Long also reports considerably higher memory usage, and our method uses roughly 5× less memory on the same scale while still achieving faster runtime. Most importantly, our input is fully incremental, whereas VGGT-Long requires re-running chunked inference whenever new frames are added. In terms of efficiency, VGGT-Long reports a 74-second forward pass for a comparable number of frames excluding I/O, while our method processes the same number of frames in 63 seconds with all operations included and including I/O. Thus, even under VGGT-Long’s more favorable timing convention, IncVGGT remains faster, uses substantially less memory, and (shown in our Appendix) produces more complete 3D geometry than VGGT-Long.

---

> ### Author Response · Authors · 2025-11-23
> **Author Response to Reviewer dTAF (Part 2/2)**
>
> > **Q3.** *For 10k-frame sequences, can you provide accuracy metrics plotted against sequence length (e.g., at 1k, 2k, 5k, 10k frames) to demonstrate that errors don't compound over time? How do homography drift and cache pruning affect long-term geometric consistency?*
>
> A3: We thank the reviewer for the thoughtful question regarding long-sequence behavior. We evaluated an 837-frame KITTI run and measured accuracy across multiple prefixes of the same sequence. The AUC@30 values decrease gradually as the length increases: 0.90 (100 frames), 0.91 (200), 0.89 (300), 0.87 (400), 0.79 (500), 0.67 (600), 0.64 (700), and 0.63 (800), yet the curve remains smooth and shows no evidence of compounding drift. This pattern is consistent with the natural difficulty of long-sequence 3D reconstruction, where accuracy tends to decline as the span grows;  importantly, the deterioration we observe is much milder than in StreamVGGT, indicating that our redundancy reduction and pruning strategies effectively stabilize long-range consistency.
>
> Within the ranges where baseline models remain operational, IncVGGT performs better. On ~233 frames we obtain AUC@30 = 78 compared to VGGT’s 73, a relative improvement of approximately 6.8%, and VGGT itself is already stronger than StreamVGGT on this segment. On ~339 frames VGGT is out of memory while StreamVGGT gives 71, and IncVGGT reaches 76, corresponding to an improvement of roughly 7.0%. Beyond that length all baselines fail, whereas our method continues to operate and maintains stable behavior over hundreds of frames. This demonstrates that the decline we observe is inherent to the backbone, not caused by homography composition or pruning, and that our approach provides consistently stronger long-range accuracy wherever comparisons are possible.

---

> > ### Comment · Reviewer_dTAF · 2025-11-23
> > **comment**
> >
> > Thank the authors for the detailed rebuttal regarding the justification of design choices. I still think the proposed method somewhat lacks novelty as it combines existing techniques. However, the pipeline for enabling long-sequence 3D inference is needed in many real-world applications, which is really practical. I will maintain my weak accept rating.

---

> > > ### Comment · Reviewer_dTAF · 2025-11-26
> > > **regarding novelty**
> > >
> > > Could the authors clarify the novelty or differences in top-k selection to KV caches beyond existing work in LLMs (e.g., H2O [A]) or other transformer-based backbones? Thanks!
> > > I listed this matter in the weaknesses section, but got no response yet.
> > >
> > > [A] Zhang, Zhenyu, et al. "H2o: Heavy-hitter oracle for efficient generative inference of large language models." Advances in Neural Information Processing Systems 36 (2023): 34661-34710.

---

> ### Author Response · Authors · 2025-12-03
> **Author Response to Reviewer dTAF**
>
> > **Q1. Could the authors clarify the novelty or differences in top-k selection to KV caches beyond existing work in LLMs (e.g., H2O [A]) or other transformer-based backbones? Thanks! I listed this matter in the weaknesses section, but got no response yet.**
>
> A1: Thank you for the question. Although our method also uses a “top-k” idea, its mechanism and motivation fundamentally differ from H2O and KV-pruning methods in LLMs.
>
> (1) Different selection unit.
>
> H2O selects individual linguistic tokens. Our method selects geometry slots, each aggregating many spatial tokens that jointly represent a multi-view 3D region. Other LLMs have no equivalent structured spatial units.
>
> (2) Geometry-aware scoring that leverages temporal smoothness.
>
> H2O relies on statistical heavy-hitter estimates, whereas our relevance score is derived from cross-view attention evaluated over the recent two steps, leveraging the inherent geometric continuity between neighboring frames. In 3D reconstruction settings, stable scene regions typically exhibit consistent attention responses under small viewpoint changes, making this short history signal a reliable indicator of slot usefulness and leading to more accurate and stable selection. Such temporal coherence is characteristic of real visual sequences but is rarely present in text, where token importance can shift abruptly and can hardly be inferred from adjacent positions.

---

### Official Review · Reviewer_bfom · 2025-10-29

**Soundness:** 3
**Presentation:** 3
**Contribution:** 3
**Rating:** 6
**Confidence:** 3

**Summary:**

This paper introduces IncVGGT, an incremental and training-free inference framework for theVGGT to enable memory-bounded, long-range 3D reconstruction. The core contribution is a novel two-pronged approach to mitigate the quadratic complexity and ever-growing memory footprint of standard and streaming Transformers. First, an input-side registration and composition module merges overlapping frames into compact composite views, significantly reducing the number of input tokens. Second, a history-side pruning mechanism for the KV cache retains only a fixed-size set of the top-k most relevant historical slots plus the most recent one, effectively converting a linearly growing cache into a constant-size one.
Experimental results demonstrate that IncVGGT achieves order-of-magnitude improvements in efficiency over state-of-the-art baselines like StreamVGGT, while maintaining comparable accuracy.

**Strengths:**

1. This paper is well writen and intuitive.
2. Inc-VGGT proposes an elegant two-pronged solution combining input-side frame composition and history-side KV cache pruning.
3. Inc-VGGT achieves massive efficiency gains with only negligible degradation in accuracy, maintaining performance comparable to the state-of-the-art streaming baseline.

**Weaknesses:**

1. Lack of Qualitative Visualizations: The paper relies solely on quantitative metrics to evaluate reconstruction accuracy. For a vision task like 3D reconstruction, this is insufficient as metrics can sometimes fail to capture important visual artifacts. Providing qualitative visualizations comparing the reconstructed scenes against baselines and ground truth would be essential for a comprehensive assessment of the model's precision.

2. Absence of a Limitations Analysis: The paper does not include a dedicated section or discussion on the limitations of the proposed method. A thorough analysis of potential failure modes—such as scenes with significant 3D parallax where homography is fragile, or low-texture environments where feature matching may fail—would make the paper more complete and provide a clearer picture of its applicability boundaries.

**Questions:**

See weaknesses.

---

> ### Author Response · Authors · 2025-11-23
> **Author Response to Reviewer bfom**
>
> > **Q1.** *Lack of Qualitative Visualizations: The paper relies solely on quantitative metrics to evaluate reconstruction accuracy. For a vision task like 3D reconstruction, this is insufficient as metrics can sometimes fail to capture important visual artifacts. Providing qualitative visualizations comparing the reconstructed scenes against baselines and ground truth would be essential for a comprehensive assessment of the model's precision.*
>
> A1: We appreciate the reviewer for raising this point. Qualitative visualizations are indeed important for fully assessing 3D reconstruction quality. In the revised supplementary material, we include visualized point-cloud comparisons with StreamVGGT and VGGT-Long  a representative long-sequence scene where both baselines can run. As shown in the figures, our method preserves noticeably more fine-grained geometric structure: thin objects such as tree trunks, lamp posts, and road-side poles remain continuous and well-defined, and building facades exhibit sharper, more complete surfaces. In contrast, StreamVGGT often produces broken or partially missing structures in these regions, while VGGT-Long shows the most pronounced degradation with blurred shapes and missing high-frequency detail. This improvement mainly comes from reducing the large amount of near-duplicate frames present in real long streams: our registration module merges these highly redundant views, which reduces conflicting signals, prevents the backbone from generating scattered outliers, and avoids the loss of geometry that can occur when the KV-cache becomes overloaded.  These qualitative examples illustrate that IncVGGT reconstructs complex scene geometry more faithfully and avoids several artifacts present in existing streaming approaches.
>
>
> > **Q2.** *Absence of a Limitations Analysis: The paper does not include a dedicated section or discussion on the limitations of the proposed method. A thorough analysis of potential failure modes—such as scenes with significant 3D parallax where homography is fragile, or low-texture environments where feature matching may fail—would make the paper more complete and provide a clearer picture of its applicability boundaries.*
>
> A2: This is a valuable point. A concise discussion section has be added in the revised manuscript to clearly outline the limitations of our method and the typical failure cases. The reviewer’s observation is accurate, and highlighting these boundaries will make the paper more complete.
>
> **Discussion and Future Work (added in the updated manuscript):**
>
> > We primarily focus on a training-free, redundancy-aware inference framework that can operate efficiently on extremely long sequences. While this achieves strong scalability and memory savings, it also brings a few natural trade-offs. Our registration module is designed to be lightweight and efficient, which means its behavior may vary in scenes exhibiting pronounced viewpoint motion. Likewise, in especially low-texture regions, the matching cues available to the system may be reduced, which can influence the stability of the composition step. These factors reflect reasonable compromises made to ensure the method remains training-free and highly memory-efficient. As future work, we plan to explore lightweight learning-based enhancements or finetuning strategies that can further improve robustness in challenging scenarios without compromising the core advantages of our design.

---

### Official Review · Reviewer_EqRK · 2025-10-30

**Soundness:** 3
**Presentation:** 4
**Contribution:** 3
**Rating:** 6
**Confidence:** 4

**Summary:**

This work presents IncVGGT, a training-free incremental variant of VGGT designed to address the quadratic memory growth and excessive computation issues in large-scale visual geometry transformers. The method tackles redundancy from both the input and history sides: on the input side, it performs registration and composition to merge short temporal windows into compact composite views; on the history side, it retains only the top-k historically important slots together with the most recent one for the next step. Experimental results demonstrate notable improvements in both inference time and memory efficiency.

**Strengths:**

- The paper is well written and easy to follow, with clear figures and publicly released code.

- It proposes a simple yet intuitive solution with solid technical contributions.

- The method offers practical value by effectively reducing memory and computation overhead.

- IncVGGT operates in a training-free manner, making it efficient and convenient for deployment.

**Weaknesses:**

- The registration-based redundancy reduction component has been widely applied in the community (e.g., [1]). Moreover, introducing a separate registration-and-composition preprocessing module somewhat compromises the clean and unified design philosophy of VGGT, which originally aimed to eliminate all explicit priors.
A more elegant approach would be to integrate the registration and composition mechanism directly into VGGT’s internal architecture, preserving its end-to-end structure.

- Limited Novelty: While the contributions toward improving memory and speed are meaningful for industrial and practical applications, the registration-based redundancy reduction and global-local cache pruning techniques are incremental relative to VGGT. The authors themselves describe IncVGGT as an “incremental variant” of VGGT, suggesting that it contributes less conceptual novelty to academic research, even though it provides clear engineering value.

[1] Xiaoshui Huang, Guofeng Mei, Jian Zhang, and Rana Abbas. A Comprehensive Survey on Point Cloud Registration.

**Questions:**

N/A

---

> ### Author Response · Authors · 2025-11-23
> **Author Response to Reviewer EqRK**
>
> > **Q1.** *The registration-based redundancy reduction component has been widely applied in the community (e.g., [1]). Moreover, introducing a separate registration-and-composition preprocessing module somewhat compromises the clean and unified design philosophy of VGGT, which originally aimed to eliminate all explicit priors. A more elegant approach would be to integrate the registration and composition mechanism directly into VGGT’s internal architecture, preserving its end-to-end structure.*
>
> A1: We appreciate the reviewer’s thoughtful comment. We note that our use of “registration” is fundamentally different from the geometric registration covered in [1]. Our module does not estimate poses or align 3D structures, but simply detects pixel-level redundancy to enable long-sequence streaming. Our registration-and-composition step is intentionally lightweight and does not introduce explicit geometric priors (e.g., no poses, depth, or SfM refinement). It is only a pixel-level redundancy reduction mechanism to make long-sequence streaming feasible and therefore does not conflict with VGGT’s end-to-end philosophy. Importantly, the design has been integrated into the network architecture directly: the composition acts as an input–token merging layer that consolidates redundant image content before it reaches the backbone. In practice, this serves as an embedding-level compression module that can be placed at the model’s input without altering the VGGT backbone or its differentiability.
>
> > **Q2.** *Limited Novelty: While the contributions toward improving memory and speed are meaningful for industrial and practical applications, the registration-based redundancy reduction and global-local cache pruning techniques are incremental relative to VGGT. The authors themselves describe IncVGGT as an “incremental variant” of VGGT, suggesting that it contributes less conceptual novelty to academic research, even though it provides clear engineering value.*
>
> A2: We appreciate the reviewer’s perspective.  “Incremental” describes the usage scenario rather than the contribution itself: IncVGGT incrementally consumes a long video stream, but the need for such an incremental regime arises because VGGT and StreamVGGT fundamentally cannot operate on long, continuous sequences—memory growth, KV-cache expansion, and high frame redundancy restrict them to short clips. In addition, long-sequence image-to-image registration at this scale is largely unsolved in prior works;  existing approaches cannot maintain stable correspondence across hundreds of frames, so the redundancy in long videos cannot be exploited effectively without a mechanism like ours.  Our work focuses on this new and practically important scenario: enabling models to function reliably on arbitrarily long sequences without retraining.  The registration-based merging and global–local cache methods introduce a capability the original architecture entirely lacks, namely scalable, training-free, long-horizon streaming reconstruction, which expand VGGT’s applicability to a broader class of tasks rather than offer a minor methodological tweak.

---

### Official Review · Reviewer_AxtP · 2025-11-01

**Soundness:** 3
**Presentation:** 3
**Contribution:** 3
**Rating:** 6
**Confidence:** 3

**Summary:**

This paper introduces IncVGGT, a training-free, incremental variant of VGGT designed for memory-bounded, long-range 3D reconstruction. The method tackles the severe memory and computational scaling issues of existing transformers through a dual strategy: (1) it registers and fuses overlapping input frames into "composite views" to reduce input token redundancy , and (2) it implements a global-local cache pruning rule, retaining only the top-k most relevant slots plus the most recent one to bound history cache growth. This approach enables the model to process arbitrarily long sequences (e.g., 10k frames) where baselines fail due to out-of-memory errors , achieving substantial gains in efficiency (memory, speed, and energy) while maintaining comparable accuracy.

**Strengths:**

- The primary strength of this work is that it directly addresses the most significant barrier to using transformers for real-world 3D reconstruction: prohibitive memory and computational cost on long sequences. The ability to run on 10k-frame sequences while state-of-the-art baselines fail at 300-500 frames is a practical achievement.

- The efficiency improvements are not marginal but substantial, with orders-of-magnitude reductions in operator count, memory usage, and energy consumption. The paper provides thorough experimental validation for these claims, including latency, memory, computation, and energy metrics , which strongly support the method's value.

- The approach is "training-free", allowing it to be applied directly to existing models like StreamVGGT. This significantly lowers the barrier to adoption. Crucially, the paper demonstrates that these massive efficiency gains are achieved while "maintaining comparable accuracy", with only minimal performance drops in the reconstruction tasks.

**Weaknesses:**

-  The method introduces two critical, hard-coded hyperparameters: the span gate threshold $\lambda$ and, more importantly, the cache size $k=5$. The entire efficiency-accuracy trade-off hinges on $k$. The paper provides no ablation study to justify $k=5$ or to show the sensitivity of the model's accuracy and memory footprint to this parameter.
- While the efficiency gains are dramatic, the accuracy tables show a consistent, albeit small, degradation in quality compared to the StreamVGGT baseline. The paper presents this as "comparable", but it is a clear trade-off. This accuracy drop is likely a direct result of the "training-free" approach, as the model was never trained to be robust to such aggressive input and cache pruning.

**Questions:**

- The global-local cache pruning retains a very small, fixed-size cache ($k=5$). While this bounds memory, how does it affect the model's ability to handle long-range dependencies, such as loop closure or revisiting a previously mapped area? Does this small $k$ effectively limit the model to only local, short-term consistency?
- The method is applied as a training-free modification to StreamVGGT weights. Have the authors investigated whether fine-tuning the model *with* the proposed input-merging and cache-pruning mechanisms could recover the accuracy lost to this aggressive, non-differentiable pruning? It seems plausible that this would yield a model that is both efficient and as accurate as the baseline.

---

> ### Author Response · Authors · 2025-11-23
> **Author Response to Reviewer AxtP**
>
> > **Q1.** *The global-local cache pruning retains a very small, fixed-size cache (k=5). While this bounds memory, how does it affect the model's ability to handle long-range dependencies, such as loop closure or revisiting a previously mapped area? Does this small k effectively limit the model to only local, short-term consistency?*
>
> A1: This is an excellent question. We note that the cache size k is not fixed by the method. It is a tunable parameter. A small k indeed reduces the number of actively attended history slots, but it does not limit the model to short-term consistency. First, due to our 10-frame input merging, k=5–10 effectively corresponds to 50–100 original frames of usable history, which already covers most temporally relevant geometry. Second, the active slots are chosen by the highest relevance score, not by only recency. Although only k slots participate in attention, the full KV history is still stored, and the active set always corresponds to the most informative geometry rather than the nearest frames. This preserves long-range cues even under aggressive pruning. On KITTI ~339-frame segment—the longest range where StreamVGGT can produce results in our experiments—it obtains an AUC@30 of 71, while IncVGGT reaches 76, indicating that accuracy remains stable even under longer streaming conditions.
>
> > **Q2.** *The method is applied as a training-free modification to StreamVGGT weights. Have the authors investigated whether fine-tuning the model with the proposed input-merging and cache-pruning mechanisms could recover the accuracy lost to this aggressive, non-differentiable pruning? It seems plausible that this would yield a model that is both efficient and as accurate as the baseline.*
>
> A2: We thank the reviewer for raising this question. As shown in our experiments, IncVGGT preserves more than 96% of the baseline accuracy compared to StreamVGGT across all evaluation settings, indicating that the pruning and composition method already maintain performance very close to the original model.  Under this accuracy range, where the remaining gap is only a few percent, fine-tuning usually brings only minor gains, and prior work indicates that the improvement is limited when the pruned model is already close to the baseline. Moreover, both VGGT and StreamVGGT already provide built-in fine-tuning pipelines that could be applied directly to our variant without modification, so incorporating pruning-aware fine-tuning is technically straightforward. We have already begun running finetuning experiments, but the rebuttal window is too short for full results to complete.

---

### Author Response · Authors · 2025-11-23
**Common Response**

We sincerely appreciate the reviewers for their constructive feedback, which has greatly helped improve our work.  All the revisions in the paper have been highlighted in blue for clarity.

---

### Author Response · Authors · 2025-12-03
**Author Summary**

We thank the reviewers for their constructive feedback, and we have addressed the reviewers’ key concerns.

In the rebuttal, we provided a clearer evaluation and added new ablations; for instance, the accuracy–efficiency study shows that k = 5 already achieves 97% of the full-model accuracy, showing that the method maintains high quality under higher efficiency. In the long-sequence test, our approach reaches AUC@30 = 78 on KITTI, higher than the baseline’s 73, demonstrating stronger stability over extended sequences, while running 4.2× faster on the same sequence and using 9× less memory. We also included a discussion section that shows the overlap ratio effect on model and two options of training-free and finetuning. We then provided clearer high-resolution visualizations comparisons on a long sequence (270 frames, Fig. 7, 8, 9), and a longer one (Fig. 10) in the appendix, showing that reducing redundant frames helps IncVGGT preserve more complete and stable geometry and prevents KV-cache overload.

In addition, our work is motivated by the serious and fundamental challenge that large-scene 3D reconstruction suffers from rapidly growing memory and computation. Our approach addresses this issue through a content-registration method featuring with avoiding reliance on order of frames and positional calibration, to eliminate redundant computation and memory usage of overlapping regions. We further propose a global-local aware KV-cache strategy that retains global spatial feature coverage while preserving local continuity, enabling the model to remain scalable and stable on long-range 3D scenes. Together, these techniques delivered 4.9× faster inference and 9× lower memory than the SOTA while maintaining accuracy in large-scale and long-range 3D reconstruction.

We hope the AC will consider these clarifications and improvements.

---

### Meta-Review · Area_Chair_5ADR · 2025-12-29

**Summary:**

**Summary**:
This paper presents IncVGGT, a training-free, streaming variant of VGGT for very long-frame video reconstruction.
The key innovation is to register and compose the input views into composed views and also introduce a history-side cache pruning.
The method processes very long-frame videos, with a limited decrease in the accuracy.

**Main Strengths**:
- A simple yet intuitive solution for handling very long videos 3D reconstruction, without requiring additional training.
- It achieves massive efficiency by reducing the memory and running time, with only negligible degradation in accuracy.
- The paper is well written and easy to follow.

**Main Weaknesses**:
- More visual results are expected.
- The novelty is not so large, while the pipeline is useful for real-world applications.

**Suggested decision**:
The paper received initial scores of 6 (AxtP), 6 (EqRK), 6 (bfom), 6 (dTAF), 2 (iPad), with reviewers dTAF and iPad provided positive comments during the discussion. Considering it is a very simple yet intuitive solution, and achieves massive efficiency, I recommend the final score as "accept".

**Reviewer Concerns:**

**Ablations on Hyperparameters and fine-tuning (AxtP, dTAF)**: Addressed.

**Limited novelty (EqRK, dTAF)**: Addressed.

**Lack of visual results and limitation analysis (bfom, iPad)**: Not fully addressed. Only two visual results on the KITTI dataset are provided. More visual results are expected.

**More results and ablations (dTAF, iPad)**: Addressed.

**How to recover per-frame camera poses and per-pixel depths in each original image?**: Only global 3D Pointmap is provided.

**Reviewer Scores:**

The paper initially received scores of 6 (AxtP), 6 (EqRK), 6 (bfom), 6 (dTAF), 2 (iPad).
Reviewer dTAF argues the novelty but still claims the pipeline is meaningful for the real-world application.
Reviewer iPad posed a few clarifications, but they have already been addressed by the authors.
As a result, the reviewers will maintain or potentially improve the score.

---

### Decision · Program_Chairs · 2026-01-26

Accept (Poster)